# Food By-Product Valorization by Using Plant-Based Coagulants Combined with AOPs for Agro-Industrial Wastewater Treatment

**DOI:** 10.3390/ijerph19074134

**Published:** 2022-03-31

**Authors:** Rita Beltrão Martins, Nuno Jorge, Marco S. Lucas, Anabela Raymundo, Ana I. R. N. A. Barros, José A. Peres

**Affiliations:** 1Centre for the Research and Technology of Agro-Environmental and Biological Sciences (CITAB)/Inov4Agro (Institute for Innovation, Capacity Building, and Sustainability of Agri-Food Production), University of Trás-os-Montes and Alto Douro (UTAD), 5000-801 Vila Real, Portugal; ritabeltraomartins@icloud.com (R.B.M.); abarros@utad.pt (A.I.R.N.A.B.); 2Centro de Química de Vila Real (CQVR), Departamento de Química, Universidade de Trás-os-Montes e Alto Douro (UTAD), Quinta de Prados, 5000-801 Vila Real, Portugal; njorge@uvigo.es (N.J.); mlucas@utad.pt (M.S.L.); 3Escuela Internacional de Doctorado (EIDO), Campus da Auga, Campus Universitário de Ourense, Universidade de Vigo, As Lagoas, 32004 Ourense, Spain; 4LEAF—Linking Landscape, Environment, Agriculture and Food, Instituto Superior de Agronomia, Universidade de Lisboa, Tapada da Ajuda, 1349-017 Lisbon, Portugal; anabraymundo@isa.ulisboa.pt

**Keywords:** circular economy, by-products reuse, plant-based coagulants, wastewater treatment, coagulation–flocculation–decantation, photo-Fenton

## Abstract

Re-using and adding value to by-products is one of the current focuses of the agri-food industry, following the Sustainable Development Goals of United Nations. In this work, the by-products of four plants, namely chestnut burr, acorn peel, olive leaf, and grape stem were used as coagulants to treat elderberry wastewater (EW), a problematic liquid effluent. EW pre-treatment using these natural coagulants showed promising results after pH and coagulant dosage optimization. However, the decrease in total organic carbon (*TOC*) was not significant, due to the addition of the plant-based natural coagulants which contain carbon content. After this pre-treatment, the photo-Fenton advanced oxidation process was selected, after preliminary assays, to improve the global performance of the EW treatment. Photo-Fenton was also optimized for the parameters of pH, H_2_O_2_, Fe^2+^, and irradiance power, and the best conditions were applied to the EW treatment. Under the best operational conditions defined in the parametric study, the combined results of coagulation–flocculation–decantation (CFD) and photo-Fenton for chestnut burr, acorn peel, olive leaf, and grape stem were, respectively, 90.2, 89.5, 91.5, and 88.7% for *TOC* removal; 88.7, 82.0, 90.2 and 93.1%, respectively, for turbidity removal; and finally, 40.6, 42.2, 45.3, and 39.1%, respectively, for TSS removal. As a final remark, it is possible to suggest that plant-based coagulants, combined with photo-Fenton, can be a promising strategy for EW treatment that simultaneously enables valorization by adding value back to food by-products.

## 1. Introduction

Currently, towards a more sustainable food system, the Food and Agriculture Organization (FAO) highlights the importance of improving efficiency along food chain production, reducing food waste, and recovering value from by-products, following the fundamentals of a circular economy [1]. In addition, several of the Sustainable Development Goals of United Nations are directly or indirectly focused on water and in its use [2]. Wastewater treatment, in particular, is an important step in the food production chain, both for water recovery and environmental impact reduction. Consequently, the use of by-products for wastewater treatment, as an environment-friendly solution, has garnered interest in recent years [3].

Food industry processing plants, even when small, need water to fabricate their final products. In addition, hygiene and food safety obligations also require large volumes of water, mainly for washing procedures. Consequently, a huge volume of wastewater is generated and must be treated according to the legal standards. The treatment of wastewater using by-products from each industry is a possibility that should be studied more deeply in a circular economy strategy [1].

Wastewater treatment can be performed through a wide range of processes; nevertheless, in order to meet legal parameters for the discharge of water into sewer systems or natural waters, it is crucial to find more sustainable and cost-effective methods [4]. To assess treated water quality, several parameters are monitored, such as chemical oxygen demand (COD), total organic carbon (*TOC*), biochemical oxygen demand (BOD_5_), total suspended solids (TSS), and turbidity [5].

Coagulation–Flocculation–Decantation (CFD) is a well-known wastewater treatment method, commonly used due to its good results. Agro-industrial wastewater contains suspended, colloidal, and dissolved particles. Although the complexity of electrostatic interactions in water is quite complex, the Stern model of the electrical double layer usually considers that colloidal particles are mostly negatively charged. CFD adds a coagulant/flocculant that will destabilize that colloidal particles, aggregating them into bigger particles (clots), followed by gravitational sedimentation and finally, decantation [6]. Most commonly used coagulants utilized in water and wastewater treatment consist of aluminum and iron salts that present several disadvantages related to the environment and public health. Since chemical coagulants are not biodegradable, traces of these compounds remain in the water after coagulation. In particular, aluminum can cause several neurodegenerative diseases, and iron is also very corrosive. Contrarily, many advantages have been found for the use of natural coagulants since they are biodegradable, non-toxic, non-corrosive, and easy to use [3,7].

Natural coagulants, mainly plants, started being used many years ago, and they are considered an ancestral technique for treating water. Recently, in developing countries, plants such as *Moringa oleifera* have been used to improve water quality in order to reduce waterborne health problems, showing effective results. Thus, when investigating the ability of natural coagulants to improve water quality, researchers began applying them in many other situations in order to discover water treatments without the drawbacks of chemical coagulants. Currently, research natural coagulants is focused on a multiplicity of organic sources, including plant leaves, flowers, or seeds, marine crustaceans or shellfish, and microorganism extracts [3,7,8].

Aiming at discovering more sustainable solutions, along with waste valorization, four different by-products from the food industry were chosen: chestnut burr, acorn peel, olive leaf, and grape stem. These by-products result from several different fruit productions: chestnut (*Castanea sativa*), acorn (*Quercus ilex* and *Quercus rotundifolia*), olive (*Olea europaea*), and grape (*Vitis vinifera*), which are abundant in many places around the world. This is an important aspect, since one of the disadvantages pointed out for natural coagulants in general is the fact that they are often only available regionally, making them more difficult to use on a large scale [7].

Several authors have been studying the use of the previously mentioned natural coagulants for treating different types of WW; however, to the best of our knowledge, in none of these studies has CFD been combined with photo-Fenton treatment [9,10,11,12,13].

According to Amor et al. [14], CFD has shown high efficiency in the pre-treatment of agro-industrial WW, reducing TSS, turbidity, and *TOC*. Ang and Mohamad [3] also recommend natural coagulant CFD treatment, in combination with other processes, in order to reach a better final result in treated WW.

Advanced Oxidation Processes (AOPs) are based on the generation of highly reactive radicals, such as hydroxyl radicals, which are capable of oxidizing organic pollutant compounds present in water. The main advantages are the possibility of chemically oxidized recalcitrant and toxic compounds with a high mineralization rate. On the other hand, specific reaction conditions and sometimes, significant treatment costs, are pointed out as AOPs limitations [14,15].

AOPs are robust WW treatment processes dependent on highly reactive free radicals, particularly hydroxyl and hydroperoxyl radicals (${\mathrm{HO}}^{•}$ and ${\mathrm{HO}}_{2}^{•}$), and radical intermediates accompanied by high efficiencies in a relatively short treatment time [16]. Among the AOPs, the Fenton reagent has been used by many researchers in WW treatment. The Fenton reagents contain an aqueous integration of hydrogen peroxide (H_2_O_2_) and ferrous ions (Fe^2+^) in an acidic medium, leading to H_2_O_2_ breaking into a hydroxyl ion and a hydroxyl radical, and the oxidation from Fe^2+^ to Fe^3+^, as observed in Equation (1), as follows [17]:
(1)$${\mathrm{Fe}}^{2+}+H_{2}O_{2}\;\to \;{\mathrm{Fe}}^{3+}+{\mathrm{HO}}^{•}+{\mathrm{HO}}^{-}$$

The photo-Fenton process (Fe^2+^/H_2_O_2_/UV) is an enhancement of the Fenton reaction that uses the UV to visible light, in a wavelength range from 250 to 600 nm, to improve the hydroxyl radical generation, as show in Equation (2) [18], and to reduce Fe^3+^ to Fe^2+^, as observed in Equations (3) and (4) [19]:(2)$$H_{2}O_{2}+\mathrm{hv}\;\to \;{2\mathrm{HO}}^{•}$$
(3)$${\mathrm{Fe}(\mathrm{HO})}^{2+}+\mathrm{hv}\;\to \;{\mathrm{Fe}}^{2+}+{\mathrm{HO}}^{•}$$
(4)$${\mathrm{Fe}}^{3+}+H_{2}O+\mathrm{hv}\;\to \;{\mathrm{Fe}}^{2+}+{\mathrm{HO}}^{•}+H^{+}$$

Actually, elderberry wastewater (EW) was chosen as an agro-industrial wastewater case study for the application of CFD with plant-based coagulants, followed by photo-Fenton, due to the fact that it is an environmental problem when released in water bodies in the Douro region in Northern Portugal. EW is characterized by presenting a high content of polyphenols, dark color, a significant amount of organic matter, and a high recalcitrant content [14,20,21].

The aim of the present work was to (1) characterize plant by-products (chestnut burr, acorn peel, olive leaf, and grape stem) and apply them as a coagulant/flocculant in EW treatment in a comparison with ferrous sulfate; (2) combine the plant by-products with the photo-Fenton (Fe^2+^/H_2_O_2_/UV-A LEDs) process, assessing its efficiency through different parameters. The treated wastewater would then be used in the irrigation of agricultural fields and gardens.

## 2. Materials and Methods

### 2.1. Reagents and Elderberry Wastewater

Ferric chloride hexahydrate (FeCl_3_·6H_2_O) was acquired from Merck (Darmstadt, Germany). Iron (II) sulfate heptahydrate (FeSO_4_·7H_2_O) was purchased from Panreac (Barcelona, Spain); hydrogen peroxide (H_2_O_2_ 30% *w*/*w*) and titanium (IV) oxysulfate solution 1.9–2.1% were bought from Sigma-Aldrich (St. Louis, MI, USA). Sodium hydroxide (NaOH) was acquired from Labkem (Barcelona, Spain) and sulfuric acid (H_2_SO_4_, 95%) from Scharlau (Barcelona, Spain).

Elderberry wastewater is generated from washing and fruit processing, and it was obtained from a company in the North of Portugal.

### 2.2. Analytical Determinations

EW characterization was performed, and the data are summarized in Table 1. The COD analysis was performed with a heating block from Hach Co. (Dusseldorf, Germany), and the colorimetric assessment was completed in a Hach DR 2400 spectrophotometer (Dusseldorf, Germany). The *TOC* determination was carried out in a Shimadzu TOC-L_CSH_ analyzer (Shimadzu, Kyoto, Japan), equipped with an ASI-L autosampler and NDIR detector, through direct injection of the filtered samples. The *TOC* was used as a sum parameter (rather than the results of sophisticated analyses such as LC/MS, GC/MS or others) to determine the overall concentration of the sum of organic compounds of a sample. *TOC* is a rapid and reliable method, as long as the results are representative for the sample composition. The hydrogen peroxide concentration was measured with a portable spectrophotometer from Hach at 410 nm, adding titanium (IV) oxysulfate (DIN 38 402H15 method). The total polyphenols content was determined following the Folin–Ciocalteau method [22]. Turbidity was assessed using a 2100N IS Turbidimeter (Hach, Loveland, CO, USA) and TSS were monitored with a portable spectrophotometer. The pH was analyzed using a 3510 pH meter (Jenway, Cole-Parmer, Cambridgeshire, UK), and electrical conductivity was measured with a condutivimeter, VWR C030 (VWR, V. Nova de Gaia, Portugal), in accordance to the methodology of the Standard Methods [23]. Iron concentration was determined by atomic absorption spectroscopy (AAS) using a Thermo Scientific™ iCE™ 3000 Series (Thermo Fisher Scientific, Waltham, MA, USA).

### 2.3. Plant-Based Coagulants Preparation

The by-products of acorn peel (*Quercus ilex* and *Quercus rotundifolia*), olive leaf (*Olea europaea*), and grape stem (*Vitis vinifera*) were provided by agro-industrial companies from the Alentejo region (South of Portugal), while chestnut burr (*Castanea sativa*) by-products were collected at the Trás-os-Montes region (North of Portugal). All materials were carried to the laboratory at the University of Trás-os-Montes and Alto Douro, Vila Real (Portugal), and dried in the oven at 70 °C for 24 h. Then, the samples were ground into powder with a groundnut miller and stored in closed plastic jars until use. Table 2 identifies in detail each by-product and Figure 1 shows the by-products before and after milling.

### 2.4. Plant-Based Coagulants Powder Characterization

In order to analyze the chemical compositions of the plant-based coagulants, an IRAffinity-1S Fourier Transform Infrared spectrometer—FTIR (Shimadzu, Kyoto, Japan) device was used, in the wavenumber range of 400–4000 cm^−1^. Samples were prepared using the KBr disc method, mixing 2 mg of sample, with 200 mg KBr, and pressing it into a mold at 10 ton cm^−2^. The microstructural characterization was carried out using scanning electron microscopy—SEM (Hitachi SEM TM 3030Plus, Tokyo, Japan).

### 2.5. Wastewater Treatment Experiments

#### 2.5.1. Coagulation–Flocculation–Decantation Experimental Set Up

The CFD experiments were performed using Jar-Test equipment (ISCO JF-4, Louisville, KY, USA), with 500 mL of EW. The system was equipped with a set of four mechanical agitators, allowing synchronized agitation. In the CFD process, four plant-based coagulants (chestnut burr, acorn peel, olive leaf, and grape stem) and one metal ion-based coagulant (ferrous sulfate, 10% *v*/*w*) were used, and the CFD process was optimized as follows: variation of EW pH (3.0, 5.0, 7.0, and 9.0) under the operational conditions: [coagulant] = 1 g L^−1^, fast mix 150 rpm for 3 min, slow mix 20 rpm for 20 min, temperature 25 °C, V = 500 mL, and sedimentation for 12 h. Next, the variation of coagulant dosage was tested (0.1, 0.5, 1.0, and 2.0 g L^−1^) under the operational conditions: pH = 3.0, fast mix 150 rpm for 3 min, slow mix 20 rpm for 20 min, temperature 25 °C, V = 500 mL, and sedimentation for 12 h. To perform decantation, the supernatant was withdrawn from a point located 2 cm below the liquid level surface in the beaker in order to determine *TOC*, turbidity, and TSS so that the effects of coagulant dose could be studied.

#### 2.5.2. Photo-Fenton Experimental Set Up

The photo-Fenton trials were performed using a self-designed lab-scale reactor (Figure 2) with 500 mL of capacity and 1.4 cm depth. The UV-A LEDs system was prepared using 12 Indium Gallium Nitride (LnGaN) LEDs lamps (Roithner AP2C1-365E LEDs) with a λ_max_ = 365 nm. Nominal consumption was 1.4 W, under 350 mA, with an optical power of 135 mW, and an opening angle of 120°. To achieve the best photo-Fenton conditions, the parameters of H_2_O_2_ concentration, pH, Fe^2+^ concentration, and irradiance power (*I*_UV_) were tested. First, the pH was tested (3.0–7.0) with the experimental conditions [H_2_O_2_] = 38.8 mM, [Fe^2+^] = 1 mM, agitation = 350 rpm, temperature = 25 °C, radiation = UV-A, *I*_UV_ = 32.7 W m^−2^, and *t* = 90 min. Next, the H_2_O_2_ concentration (9.7–77.6 mM) was tested under the respectively experimental conditions of pH = 3.0, [Fe^2+^] = 1.0 mM, agitation = 350 rpm, temperature = 25 °C, radiation = UV-A, *I*_UV_ = 32.7 Wm^−2^, and *t* = 90 min. Then, Fe^2+^ concentration (0.5–5 mM) was tested under the experimental conditions: pH = 3.0, [H_2_O_2_] = 38.8 mM, agitation = 350 rpm, temperature = 25 °C, radiation = UV-A, *I*_UV_ = 32.7 Wm^−2^, and *t* = 90 min; Finally, testing *I*_UV_ (0.0–32.7 Wm^−2^) was completed, with the experimental conditions of: pH = 3.0, [H_2_O_2_] = 38.8mM, [Fe^2+^] = 1.0 mM, agitation = 350 rpm, temperature = 25 °C, and *t* = 90 min. The photo-Fenton process was stopped using catalase.

### 2.6. Kinetic Analysis

The kinetic study was based on the evolution of *TOC*, the concentration of total organic carbon dissolved in EW (mg C L^−1^). In all experiments performed, the results obtained revealed that *TOC* reduction followed a pseudo-first-order kinetics (Equation (5)):(5)$$r=-\frac{dTOC}{\mathrm{dt}}=k\;TOC\;$$
where *r* is the reaction rate, at time *t* and *k* is the pseudo-first-order kinetic constant. This equation can be integrated between times *t* = 0 and *t* = *t* to yield (Equation (6)):(6)$$ln\frac{TOC}{TOC_{0}}=-kt$$

According to this expression, a plot of the first term vs. time yield a straight line satisfying Equation (6), where the slope is *k* (min^−1^), obtaining (Equation (7)) [24]:(7)$$TOC=TOC_{0}\cdot e^{-kt}$$

The removal rate X_i_ is defined as the percentage of a specific EW parameter (turbidity, TSS, *TOC*, COD, and total polyphenols) that is reached with the wastewater treatment. X_i_ can be determined in accordance to Equation (8), as follows [25]:(8)$$X_{i}\;(\%)=\frac{C_{0}-C_{f}}{C_{0}}*100$$
where C_0_ and C_f_ are, respectively, the initial and final concentrations of parameter i.

### 2.7. Statistical Analysis

All the experiments were performed at least in duplicate. Statistical analysis was performed with OriginPro 2019 software (Originlab, Northampton, MA, USA). The statistical tool chosen to compare experimental data was the analysis of variance (one-way ANOVA). In order to compare mean values, the Tukey test was used at a confidence level of 95%, with differences between mean values being considered statistically significant at (*p <* 0.05). The differences between values are represented in the text by different letters (a, b, c, d). All data were presented as mean value and respective standard error.

## 3. Results and Discussion

### 3.1. Plant-Based Coagulants Powder Characterization

The chemical structure of a coagulant is an important aspect of its efficiency in the CFD process. Thus, the samples were analyzed through FTIR with the aim of assessing which functional groups characterize the four plant-based coagulants used in the present work. In Figure 3, FTIR spectra are represented where it is possible to observe the different bands detected, which were quite similar among the four coagulants.

The first bands were within the range from 3400 to 3250 cm^−1^, representing the stretching vibration of the O-H bond of the hydroxyl functional group (OH) that is present in proteins, fatty acids, carbohydrates, cellulose, and lignin [26,27,28,29,30]. Furthermore, the band at 2900 cm^−1^ identified the C–H stretching vibrations of CH_3_ and CH_2_ from aliphatic structures attributed to fatty acids, lipids, protein, and polyphenols [26,27,28,29,31]. The band range between 1800 and 1700 cm^−1^ is ascribed to the C=O stretching vibrations in the carbonyl groups. The vibration bands between 1650–1540 cm^−1^ were assigned to the N-H bond from the amines group and the 1250 cm^−1^ vibration band indicates the C-O stretching vibration of the phenols [26,27,28,29,30,31]. As expected, lignin presence is confirmed through the band range of 1530–1508 cm^−1^, characteristic of lignin aromatic skeletal vibrations [27,28]. In addition, the band around 1050 cm^−1^ is assigned to the C–O bond from the lignin structure [28,29]. The band within the range between 1100 and 950 cm^−1^ indicates the presence of polysaccharides [27,28]. Finally, the band at 800 cm^−1^ (approximately) could be assign to the presence of the C–H bond belonging to the structure of cellulose [28].

From this spectra analysis, it is possible to determine that plant-based coagulants are a complex matrix with different functional groups such as fatty acids, proteins, and polysaccharides, important components for their coagulation skills [32].

In Figure 4, SEM images of each coagulant (chestnut burr, acorn peel, olive leaf, and grape stem) are represented, and it is possible to observe their microstructures. Plants in general possess naturally formed porous structures which increase their contact surface area, leading to a larger electroactive surface, and consequently, to a more efficient electron transfer [32]. Important, gaps between structures could also allow for a better adsorption process, since these structures offer a vast contact surface area [33].

### 3.2. Coagulation–Flocculation–Decantation Experiments

EW retains negative colloids in suspension that are destabilized when they come into contact with positive charges of cationic proteins from coagulants. Once destabilized, these particles start aggregating and form flocs that then settle by gravity [34]. In a previous section, the FTIR spectra of plant-based coagulants have exhibited proteins in their structures, which are organic polymers, developing the role of natural polyelectrolytes. The amino acids in contact with the WW suffer an ionization reaction, producing carboxylate ions and protons. Since the colloids in suspension present a negative charge, they will be attracted by the protons, establishing a neutral molecule that will then flocculate and be aggregate in bigger particles [6].

CFD is a complex mechanism, combining the adsorption and charge neutralization with adsorption and interparticle bridging, finally leading to the precipitation of the flocs [34]. The coagulant surface charge promotes adsorption, while the release of anionic functional groups from the coagulant supports the coagulation process [9]. The most important conditions affecting CFD efficiency are pH, the type of coagulant, and its dosage [34].

To maximize CFD efficiency, the conditions of pH and coagulant dosage were optimized in relation to *TOC*, turbidity, and TSS removal. Turbidity removal is a crucial step for the success of CFD as a pre-treatment; since it is followed by the photo-Fenton process, the penetration of light into the WW is essential for triggering the photocatalytic process [35].

#### 3.2.1. Effect of pH

To determine the best conditions for performing the CFD process, the EW pH was set between 3.0 and 9.0, using conditions previously mentioned in Section 2.5.1.

In Figure 5, a higher efficiency for plant-based coagulants is observed at pH 3.0, and for ferrous sulfate at pH 5.0. The results show a turbidity removal of 58.0, 43.6, 43.5, 51.9, and 69.6% (Figure 5a) and a TSS removal of 27.0, 23.4, 20.7, 23.8, and 5.4% (Figure 5b), respectively, for chestnut burr, acorn peel, olive leaf, grape stem, and ferrous sulfate (used as the conventional coagulant control experiment). Regarding *TOC* removal, 0.0% was observed for plant-based coagulants and 2.8% for ferrous sulfate (results not shown). These *TOC* values result from the fact that plant-based coagulants are carbon compounds, which become more difficult to reduce the organic matter.

The pH influences the behavior of the coagulants, as molecules can accept or donate protons, depending on the pH value [34]. The efficiency of natural coagulants depends on pH, since the coagulation process is based on positive and negative charges, which pH is responsible for changing. Thus, when comparing the obtained results with the results of other authors, several different pH conditions were found, particularly in regards to WW and coagulant characteristics, to lead to distinctive pH optimum operational conditions [9,10,13]. Regarding ferrous sulfate, pH 5.0 showed a higher efficiency, since colloidal particles are negatively charged when the pH is around this value, contributing to charge neutralization, interparticle bridging, precipitation, and enmeshment [36].

#### 3.2.2. Dosage Effect

In this section, the coagulant dosage optimization was performed using conditions previously mentioned in Section 2.5.1. The results in Figure 6 showed a higher efficiency with the application of 0.1, 0.5, 0.5, 0.5, and 0.1 g L^−1^, respectively, for chestnut burr, acorn peel, olive leaf, grape stem, and ferrous sulfate. Using the best coagulant dosage, the following results were obtained: (1) a turbidity removal of 85.8, 82.0, 76.0, 81.9, and 74.9%, (2) a TSS removal of 35.5, 30.5, 26.6, 33.6, and 6.3%, and (3) a *TOC* removal of 0.0, 15.2, 0.0, 0.0, and 20.1%, respectively, for chestnut burr, acorn peel, olive leaf, grape stem, and ferrous sulfate. As previously seen in SEM images, the coagulants microstructure exhibited rough surfaces, porosity, and chainlike areas, which, according to Vunain et al. [37], promotes ion adsorption, and consequently contributes to reduce turbidity in WW.

Regarding plant-based coagulants, chestnut burr presented the lowest dosage in comparison with the other organic coagulants, which could be explained by its microstructure characteristics being more effective in the coagulation process.

In regards to the plant-based coagulant results, above the ideal dosage, a reduction of turbidity and TSS removal was observed (Figure 6). This can be explained due to the increase in carbon released into the EW, since a higher dosage releases a higher amount of organic matter, also affecting the charge interactions, leading to a decrease in the CFD efficiency [13]. Concerning ferrous sulfate, above the optimum dosage, the excess of positive charges led to the re-stabilization of colloidal particles as a result of charge repulsion due to the presence of positive ions, which led to a decrease in turbidity and *TOC* removal [34].

Fawzy et al. [12] analyzed the adsorption efficiency of olive leaf powder, concluding that an increasing dosage beyond the optimal performance had a negative effect, owing to the blockage of the binding area, a reduction in ions interactions, and the disruption of the surface charges. In a study on the effects of natural coagulants obtained from *Castanea sativa* and *Quercus robur* on water turbidity removal, it was also found that a lower dosage revealed better coagulation effects. These authors obtained 80% and 70% of turbidity removal for chestnut and oak based coagulants, respectively; these results are quite similar to those in the present study [13]. Comparing coagulants dosage with other authors, Kuppusamy et al. [9] used *Quercus robur* (acorn peel) to remove dyes from WW, and applied a dosage of 5 g L^−1^. The effect of olive leaf powder was studied in removing crude oil from water, with the best results obtained with a dosage of 3 g L^−1^ [10]. The results were different from the ones obtained in the present research; however, coagulation is a complex process, and the dosage effect is dependent on the type of coagulant, water pH, characteristics of WW, and other coagulation conditions [11,13,34].

### 3.3. Photo-Fenton Experiments

#### 3.3.1. Chemical Degradability of Elderberry Wastewater

In a previous section, the EW was treated by the CFD process. This treatment revealed a high efficiency in the removal of turbidity and TSS; however, a low percentage of *TOC* was removed. The reduced *TOC* removal may be due to the difficulty in the degradation of organic compounds such as polyphenols. Thus, advanced oxidation processes (AOP) were assessed in order to improve the global performance of the EW treatment.

Figure 7 shows the results obtained for the EW chemical degradability used to evaluate the efficiency and benefit of each AOP. The following processes were evaluated: (1) H_2_O_2_, (2) UV-A radiation, (3) Fe^2+^ + UV-A, (4) H_2_O_2_ + UV-A, (5) H_2_O_2_ + Fe^2+^ (Fenton), and (6) H_2_O_2_ + Fe^2+^ + UV-A (photo-Fenton). It can be observed that UV-A and H_2_O_2_ did not have any significant effects on *TOC* removal within 90 min. The H_2_O_2_ + UV-A and Fe^2+^ + UV-A processes showed a low *TOC* removal (12.1 and 13.5%, respectively) after a reaction time of 90 min. Finally, the combination of H_2_O_2_ with Fe^2+^ (Fenton reagent) and H_2_O_2_ + Fe^2+^ + UV-A (photo-Fenton process) showed 45 and 88% *TOC* removal, respectively, after 90 min.

Therefore, it is possible to deduce that the most efficient treatment of EW was the photo-Fenton process. In the present work, UV-A LEDs were used to test an irradiation source with a wavelength near visible light. This was a cheaper and more environmentally friendly irradiation selection in comparison to UV-C mercury vapor lamps.

The reason why photo-Fenton process was more efficient (88%) in *TOC* removal is explained by the higher generation of hydroxyl radicals ${\mathrm{HO}}^{•}$ when compared with Fenton reagent (45%), as previously shown by Equations (2)–(4). The results have shown that UV radiation fills as important role regarding organic matter degradation when compared with the Fenton process due to the regeneration of Fe^2+^, also reducing the amount of iron sludge formed [38].

#### 3.3.2. Effect of pH

The effect of pH on the EW treatment by the photo-Fenton process was evaluated (Figure 8). In order to assess pH influence, four different values were tested (3.0–7.0), under the following operational conditions: [H_2_O_2_] = 38.8 mM, [Fe^2+^] = 1 mM, agitation = 350 rpm, temperature = 25 °C, radiation = UV-A, *I*_UV_ = 32.7 W m^−2^, and *t* = 90 min. The results presented in Figure 8 show a *TOC* removal of 88.1, 74.8, 83.7, and 86.6%, respectively, for pH 3.0, 4.0, 6.0, and 7.0.

Regarding Appendix A, the maximum H_2_O_2_ consumption was observed at pH 3.0 (49.0%), suggesting a higher amount of hydroxyl radical generation. This higher H_2_O_2_ consumption can be explained by the iron speciation diagram as a function of pH. In fact, as observed in Appendix A, the Fe^2+^ concentration after 90 min was 7.66, 5.40, 4.84, and 1.45 mg Fe L^−1^, respectively, for pH 3.0, 4.0, 6.0, and 7.0. Thus, by increasing the pH, the iron precipitation as iron hydroxide Fe[OH]_3_ takes place, decreasing the ${\mathrm{HO}}^{•}\;$radicals available in the medium and diminishing the *TOC* removal [35,39,40]. Therefore, pH 3.0 was selected as the best operational pH value.

#### 3.3.3. Effect of H_2_O_2_ Concentration

The effect of H_2_O_2_ concentration in the EW treatment using the photo-Fenton process was evaluated (Figure 9). The H_2_O_2_ concentration ranged between 9.7 and 77.6 mM, under the following experimental conditions: pH = 3.0, [Fe^2+^] = 1.0 mM, agitation = 350 rpm, temperature = 25 °C, radiation = UV-A, *I*_UV_ = 32.7 Wm^−2^, and *t* = 90 min. Figure 9 shows the *TOC* removal rates of 48.6, 62.6, 88.1, and 75.9%, respectively, for 9.7, 19.4, 38.8, and 77.6 mM H_2_O_2_. Regarding Table 3, it is possible to conclude that with the application of 9.7 and 19.4 of H_2_O_2_, a similar H_2_O_2_ consumption (44.7 and 45.3%, respectively), was observed, which increased to 59% upon adding 38.8 mM of H_2_O_2_. However, when adding 77.6 mM of H_2_O_2_, *TOC* removal decreased, indicating that the ${\mathrm{HO}}^{•}$ radical production was lower. Similar results have been found in other studies, where the concentration of H_2_O_2_ also revealed a threshold value, for which the effect was the opposite, and the TOC removal rate decreased [21,39]. Although increasing the H_2_O_2_ concentration produced more ${\mathrm{HO}}^{•}$ radicals, the quenching effect of H_2_O_2_ occurred, decreasing the photo-Fenton’s efficiency, as observed in Equations (9) and (10), as follows [41]:(9)$$H_{2}O_{2}+{\mathrm{HO}}^{•}\;\to \;H_{2}O+{\mathrm{HO}}_{2}^{•}$$
(10)$${\mathrm{HO}}_{2}^{•}+{\mathrm{HO}}^{•}\;\to \;H_{2}O+O_{2}$$

Consequently, according to these results, 38.8 mM H_2_O_2_ was selected as the best performing concentration for the EW photo-Fenton treatment.

#### 3.3.4. Effect of Fe^2+^ Concentration

In this section, the Fe^2+^ concentration effect in the photo-Fenton EW treatment was studied. The Fe^2+^ concentration varied from 0.5 mM to 5 mM, under the following experimental conditions: pH = 3.0, [H_2_O_2_] = 38.8 mM, agitation = 350 rpm, temperature = 25 °C, radiation = UV-A, *I*_UV_ = 32.7 Wm^−2^, and *t* = 90 min. From Figure 10, it was possible to observe a *TOC* removal rate of 75.3, 88.1, 79.8, and 66.6%, respectively, for 0.5, 1.0, 2.5, and 5.0 mM of Fe^2+^. Regarding Table 3, H_2_O_2_ consumption values were 28.5, 49.0, 38.5, and 44.7%, respectively. Clearly, with the application of 0.5 mM Fe^2+^, the catalyst concentration was insufficient to generate ${\mathrm{HO}}^{•}$ and degrade the organic matter; however, 1.0 mM Fe^2+^ showed the highest H_2_O_2_ consumption, revealing higher levels of hydroxyl radical formation. When increasing the Fe^2+^ concentration above this optimum value, *TOC* removal decreased. These results were in accordance with Rodríguez-Chueca et al. [39], who applied increasing iron concentrations, and observed that photo-Fenton reaction efficiency also increased, but above the optimum iron level, the extension of the reaction declined, or scavenger effects occurred. Moreover, this study revealed that in order to avoid these phenomena, it is important to test and find the lowest possible ratio of Fe/H_2_O_2_, reducing the potential Fe and H_2_O_2_ recombination, and lowering the production of iron complexes.

It also important to note that increasing the iron amount not only reveals an undesirable effect concerning *TOC* removal, but also increases residual iron, as can be seen in Appendix A. With the application of 5.0 mM, the residual iron was much higher than with 1.0 mM (7.66 and 30.81 mg Fe L^−1^, respectively). The high concentration of Fe^2+^ ions reacted to the produced ${\mathrm{HO}}^{•}$ radicals as a radical scavenger, according to Equation (11), as follows [42]:(11)$${\mathrm{Fe}}^{2+}+{\mathrm{HO}}^{•}\;\to \;{\mathrm{Fe}}^{3+}+{\mathrm{HO}}^{-}$$

In accordance to the obtained results, 1.0 mM Fe^2+^ was selected as the best concentration for the treatment of EW using the photo-Fenton process.

A pseudo-first order, instead of a second order, reaction was assumed, due to the experimental data obtained. The hydroxyl radicals generated reacted almost instantly with the organic compounds, making its concentration levels virtually constant.

#### 3.3.5. Effect of Irradiance Power (*I*_UV_)

Finally, the UV-A radiation intensity was tested, ranging from 0.0 to 32.7 Wm^−2^, in order to determine which would be the most efficient intensity for the EW photo-Fenton treatment, under the following experimental conditions: pH = 3.0, [H_2_O_2_] = 38.8 mM, [Fe^2+^] = 1.0 mM, agitation = 350 rpm, temperature = 25 °C, and *t* = 90 min. From the results depicted in Figure 11, it was possible to observe a *TOC* removal of 45.9, 58.5, 71.5, and 88.1%, respectively, for 0.0, 5.2, 18.3, and 32.7 W m^−2^, showing that the highest radiation intensity lead to the highest *TOC* removal. According to Table 3, H_2_O_2_ consumption augmented with the increase in irradiance power 0.0, 28.2, 39.6, and 49.0%, respectively. Without application of UV-A radiation (*I*_UV_ = 0.0 W m^−2^), the homolysis of H_2_O_2_ into ${\mathrm{HO}}^{•}$ radicals was dependent on the catalyst present in the solution; however, as expect from Equation (2), with the application of radiation, a higher ${\mathrm{HO}}^{•}$ radical production took place. On the other hand, the application of radiation allowed a higher conversion of Fe^3+^ to Fe^2+^ (Equation (4)), leading to a higher *TOC* removal. Accordingly, Rodríguez-Chueca et al. [43] also observed that boosting the radiation power improved the results, increasing organic matter removal.

Thus, considering these results, 32.7 W m^−2^ was chosen as the best irradiance power to use in the EW photo-Fenton treatment.

Considering that the photo-Fenton process follows a pseudo first-order kinetic rate, the synergetic effect (S) can be calculated from Equation (12), as follows [44]:(12)$$S=(1-\frac{{(k}_{H_{2}O_{2}+\mathrm{UV}\text{-}A}{+k}_{{\mathrm{Fe}}^{2+}+\mathrm{UV}\text{-}A}{+k}_{H_{2}O_{2}{+\mathrm{Fe}}^{2+}})}{k_{H_{2}O_{2}{+\mathrm{Fe}}^{2+}+\mathrm{UV}\text{-}A}})\times 100$$
where $k_{H_{2}O_{2}+\mathrm{UV}\text{-}A}$, $k_{{\mathrm{Fe}}^{2+}+\mathrm{UV}\text{-}A}$, $k_{H_{2}O_{2}{+\mathrm{Fe}}^{2+}}$ and $k_{H_{2}O_{2}{+\mathrm{Fe}}^{2+}+\mathrm{UV}\text{-}A}$ are the pseudo first-order kinetic constants (for S > 0—synergistic effect, for S = 0—cumulative effect, for S < 0—antagonistic effect).

The results in Table 3 showed a growth of the synergistic effect with the increase in irradiance power (36.4 and 64.4%, respectively for 18.3 and 32.7 W m^−2^), except for 5.2 W m^−2^. These results revealed that increasing the irradiance power (18.3 and 32.7 W m^−2^) allowed for a higher production of ${\mathrm{HO}}^{•}$, thereby increasing the kinetic rate of *TOC* removal. Thus, the synergy results show that simultaneously using the three components (H_2_O_2_/Fe^2+^/UV-A) of photo-Fenton led to better results than the sum of its individual parts (H_2_O_2_/UV-A + Fe^2+^/UV-A + H_2_O_2_/Fe^2+^), indicating a genuine synergistic effect. In the Appendix A, Appendix A shows pseudo-first order kinetic constants, half-lives, and correlation coefficients (R^2^) along the photo-Fenton processes under different operational conditions. Appendix A also present the respective theoretical fitting curves.

### 3.4. Combination of Coagulation–Flocculation–Decantation and UV-A-Fenton

The CFD process using the four plant-based coagulants (chestnut, acorn, olive, and grape) and ferrous sulfate was optimized with the objective of determining the best EW treatment results (Section 3.2). Next, UV-A-Fenton was also optimized (Section 3.3), achieving the following best operational conditions: [H_2_O_2_] = 38.8 m, [Fe^2+^] = 1.0 mM, pH = 3.0, *I*_UV_ = 32.7 W m^−2^, agitation = 350 rpm, temperature = 25 °C, and *t* = 90 min.

In this section, CFD treatment was combined with the photo-Fenton process to achieve a better final result for EW treatment, specifically in regards to *TOC* removal, since as presented previously, plant-based coagulants are carbon compounds, which makes it more difficult to reduce organic matter using exclusively CFD without further treatment [45].

Figure 12 shows the results concerning *TOC* removal using the combined treatment of CFD (after 12 h of decantation/sedimentation) and the photo-Fenton process. Analyzing each coagulant, *TOC* removal rates at the end of EW treatment were respectively: 90.2% for chestnut burr, 89.5% for acorn peel, 91.5% for olive leaf, 88.7% for grape stem, and 83.1% for ferrous sulfate. These results show the efficiency of plant-based coagulants in EW treatment, particularly as a pre-treatment before combination with photo-Fenton. It is possible to conclude that olive leaves have shown the best results; nevertheless, each of the plant-based coagulants showed a similar effect on the *TOC* removal rate.

The application of the CFD process prior to the photo-Fenton treatment improved its efficiency, increasing the *TOC* removal rate from EW, due to an increase in the removal rates for turbidity and TSS. Moreover, the use of CFD as a pre-treatment also revealed a reduction in H_2_O_2_ consumption (49.0, 36.5, 36.4, 36.9, 36.6, and 36.8%, respectively). Therefore, the CFD process not only improved photo-Fenton performance, but also decreased the amount of reagent used. For ferrous sulfate, *TOC* removal was lower than in the photo-Fenton method without the use of the CFD process (Figure 11). This could be explained due to the excess of iron in the solution that consumed the ${\mathrm{HO}}^{•}$ radicals, decreasing the efficiency of the photo-Fenton process, as observed in Equation (11).

The pre-treatment of the EW using the CFD process was essential to reduce turbidity and TSS, improving the efficiency of the photo-Fenton process. The compounds used in the CFD process are considered to be hydroxyl radical scavengers; it is also important to remove suspended solids, as well as dark color, for better penetration of UV-A radiation, with the main objective of triggering the photo-Fenton process [35].

The results in Figure 13 show high turbidity and TSS removal rates with CFD performance, which may have improved the photo-Fenton process, allowing it to achieve higher organic carbon removal. After combining CFD with the photo-Fenton process, a significant increase in the turbidity removal rates up to 88.7, 82.0, 90.2, 93.1, and 87.7%, was observed for chestnut burr, acorn peel, olive leaf, grape stem, and ferrous sulfate, respectively, and an increase in the TSS removal rate up to 40.6, 42.2, 45.3, 39.1, and 43.4%, respectively.

The removal of polyphenols is also an important aspect, since they are responsible for EW color, and also are a cause of toxicity for the environment [46]. As it is possible to observe in Figure 13, CFD showed almost no effect on polyphenols concentration. Nevertheless, after performing the photo-Fenton treatment, the total removal of polyphenols content increased to more than 99.5%. According to Lucas and Peres [46], the oxidation of polyphenols was much faster when compared with the degradation of organic carbon, which could explain the nearly 100% polyphenols removal rate after performing the photo-Fenton process.

## 4. Conclusions

The by-products chestnut burr (*Castanea sativa*), acorn peel (*Quercus ilex* and *Quercus rotundifolia*), olive leaf (*Olea europaea*), and grape stem (*Vitis vinifera*) were used as plant-based coagulants for the CFD pre-treatment of elderberry wastewater, followed by treatment with the UV-A-Fenton process. Considering the results, it is possible to conclude that:The plant-based coagulants are a carbon-based material with proteins, fatty acids, carbohydrates, cellulose, and lignin in their constitution.The plant-based coagulants achieve higher efficiency at pH 3.0. The EW pre-treatment using the CFD process is important to reduce turbidity and total suspended solids in order to improve the efficiency of the photo-Fenton process.Under the best operational conditions, the photo-Fenton process achieves 88.1% *TOC* removal, and the synergistic effect is more evident with the application of higher irradiance power.The performance of pre-treatment using the CFD process increases the efficiency of the photo-Fenton process, with higher *TOC* removal rates.

The use of plant-based coagulants is a promising strategy for the treatment of wastewater, while at the same time, adding value to the by-products from the agri-food industry.

## Figures and Tables

**Figure 1 ijerph-19-04134-f001:**
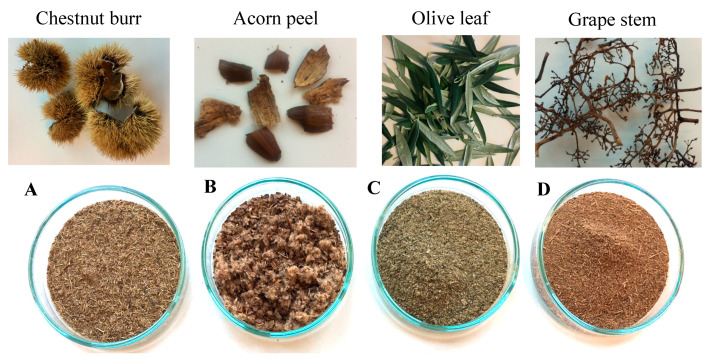
By-products used as plant-based coagulants and their respective powders: (**A**)—chestnut burr, (**B**)—acorn peel, (**C**)—olive leaf, (**D**)—grape stem.

**Figure 2 ijerph-19-04134-f002:**
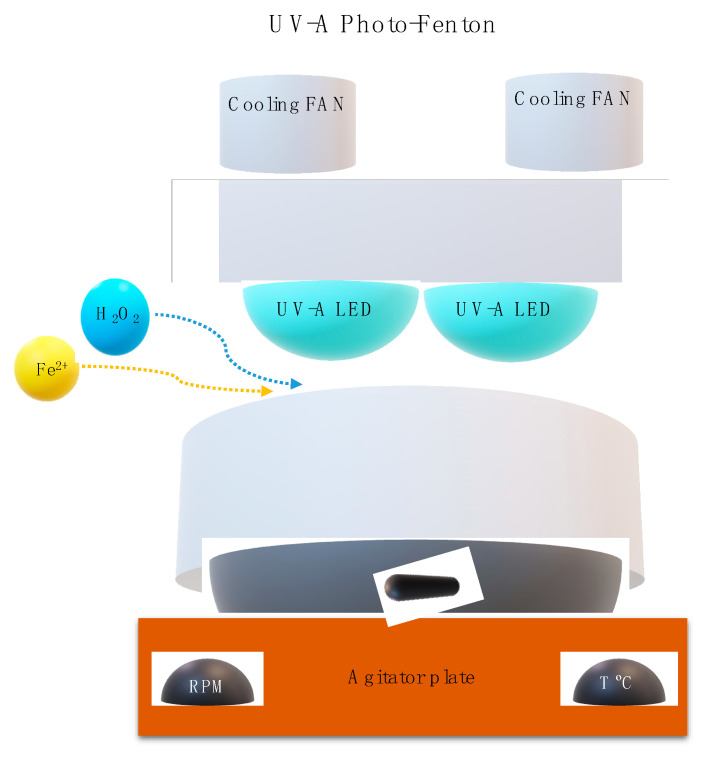
Photo-Fenton set-up schematic representation.

**Figure 3 ijerph-19-04134-f003:**
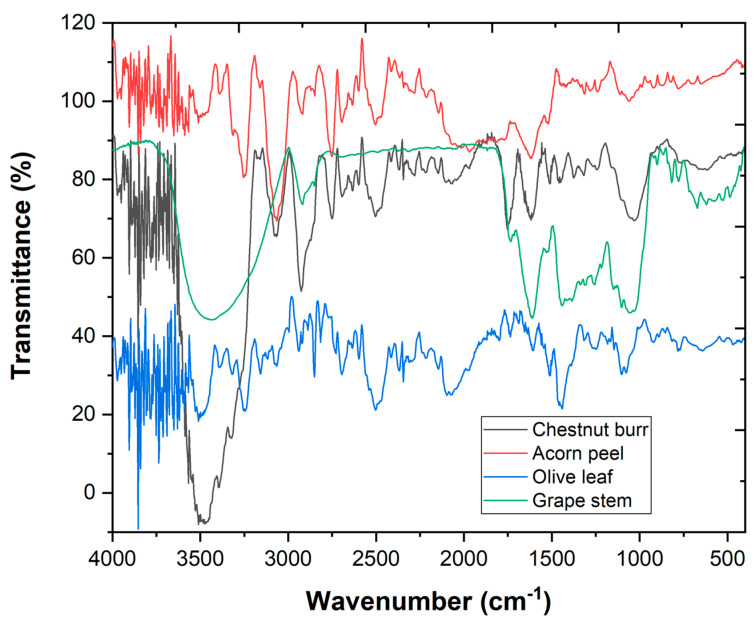
FTIR spectra of the four plant-based coagulants.

**Figure 4 ijerph-19-04134-f004:**
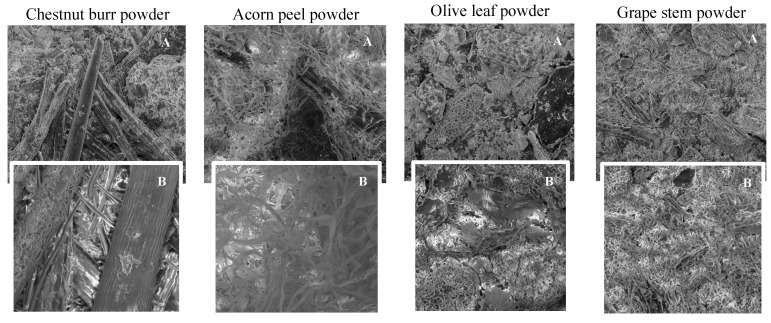
SEM images of each plant-based coagulant identified in the figure (**A**)—60× magnification and (**B**)—100× magnification.

**Figure 5 ijerph-19-04134-f005:**
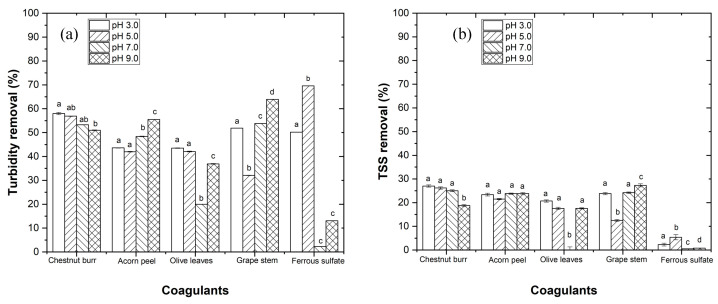
Progress of (**a**) turbidity and (**b**) TSS removal, testing different pH values (3–9). Experimental conditions: 1.0 g L^−1^ for each coagulant, *T* = 25 °C, fast mix 150 rpm for 3 min, slow mix 20 rpm for 20 min, sedimentation time of 12 h ([TOC]_0_ = 144 mg C L^−1^, turbidity_0_ = 16 NTU, TSS_0_ = 64 mg L^−1^). Different letters between the graphic bars mean that values are statistically different (*p* < 0.05). When bars have two letters (ab), it means the values are not statistically different from a and from b.

**Figure 6 ijerph-19-04134-f006:**
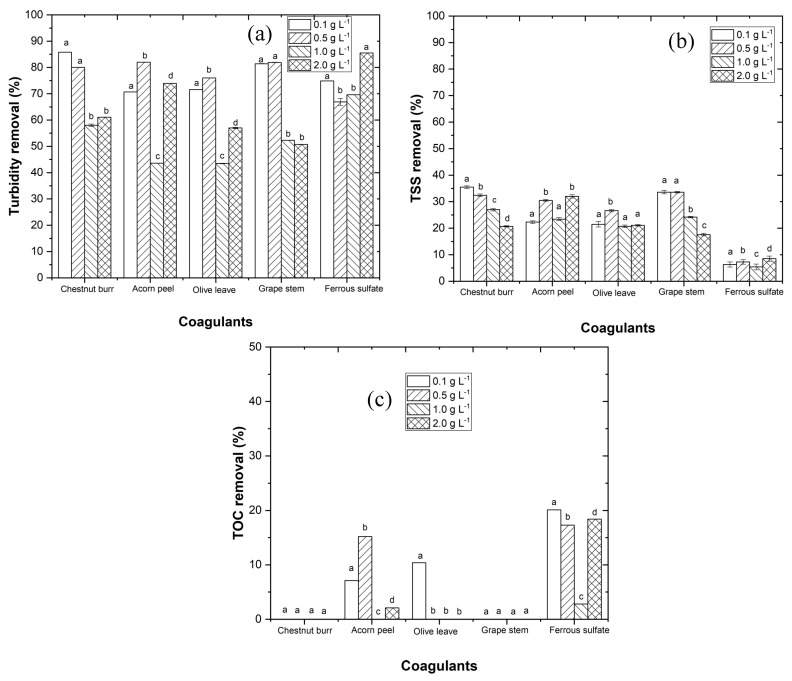
Progress of (**a**) turbidity, (**b**) TSS and (**c**) *TOC* removal along the CFD process, testing different coagulant dosage (0.1, 0.5, 1.0, 2.0 g L^−1^). Experiment conditions: pH = 3 (for plant-based coagulants), pH = 5 (for ferrous sulfate), *T* = 25 °C, fast mix 150 rpm for 3 min, slow mix 20 rpm for 20 min, sedimentation time of 12 h ([TOC]_0_ = 144 mg C L^−1^, turbidity_0_ = 16 NTU, TSS_0_ = 64 mg L^−1^. Different letters between the graphic bars mean that values are statistically different (*p* < 0.05).

**Figure 7 ijerph-19-04134-f007:**
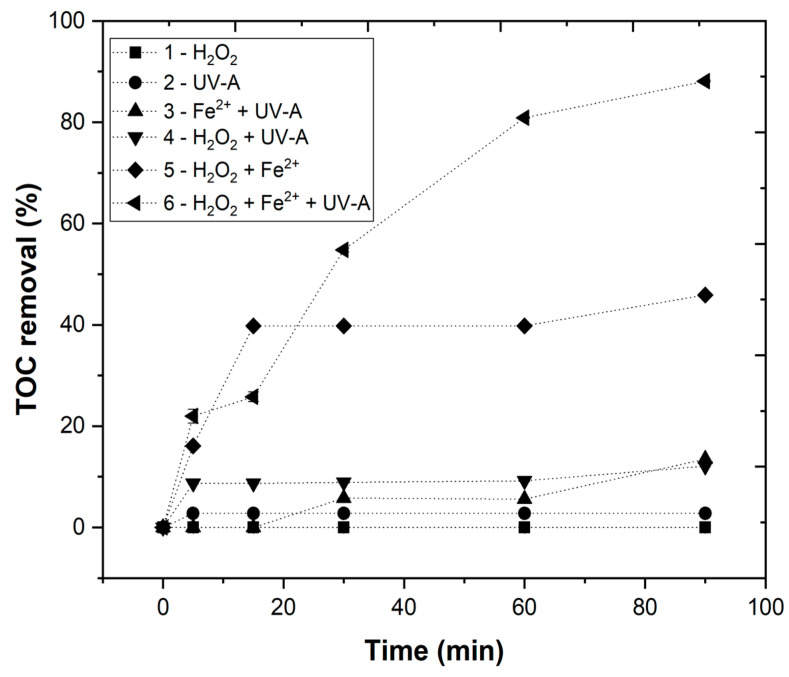
Chemical degradability of EW (*TOC* % removal) under different experimental conditions: [H_2_O_2_] = 38.8 mM, [Fe^2+^] = 1.0 mM, pH = 3.0, agitation = 350 rpm, *T* = 25 °C, radiation = UV-A, *I_UV_* = 32.7 Wm^−2^, *t* = 90 min.

**Figure 8 ijerph-19-04134-f008:**
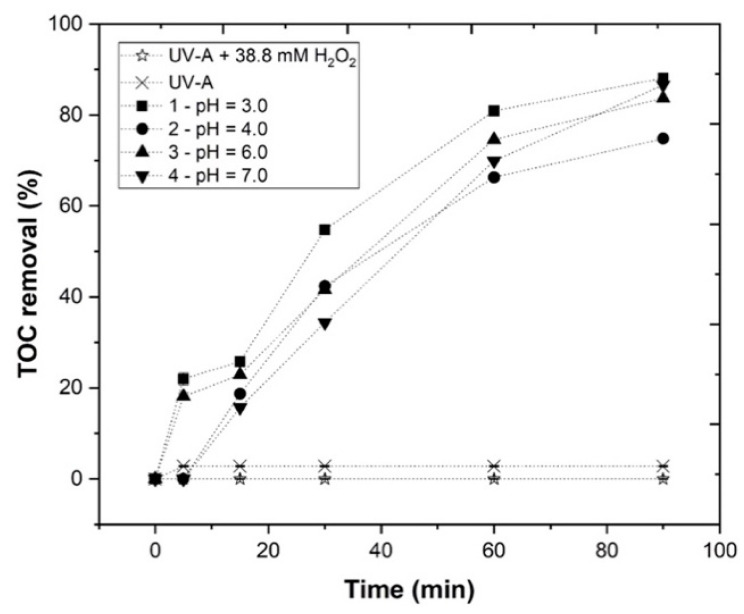
Progress of *TOC* percentage removal at different pH (3–7) in the UV-A-Fenton process. Experimental conditions: [H_2_O_2_] = 38.8 mM, [Fe^2+^] = 1.0 mM, agitation = 350 rpm, *T* = 25 °C, radiation = UV-A, *I*_UV_ = 32.7 Wm^−2^, *t* = 90 min. Blank non-catalytic experiments (UV-A/H_2_O_2_ and UV-A without H_2_O_2_) are also represented as a reference.

**Figure 9 ijerph-19-04134-f009:**
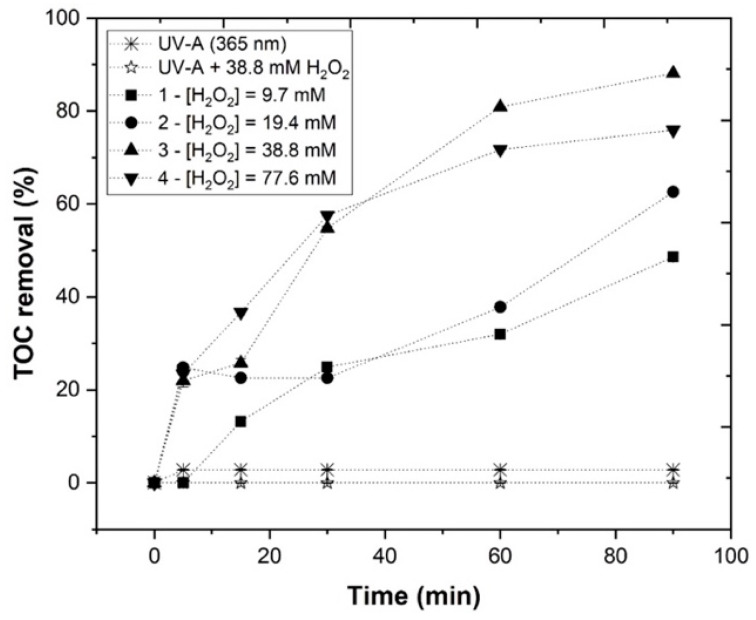
Progress of *TOC* percentage removal at different H_2_O_2_ concentrations (9.7–77.6 mM) in the UV-A-Fenton process. Experimental conditions: [Fe^2+^] = 1.0 mM, pH = 3.0, agitation = 350 rpm, *T* = 25 °C, radiation = UV-A, *I*_UV_ = 32.7 Wm^−2^, and *t* = 90 min. Blank non-catalytic experiments (UV-A/H_2_O_2_ and UV-A without H_2_O_2_) are also represented as a reference.

**Figure 10 ijerph-19-04134-f010:**
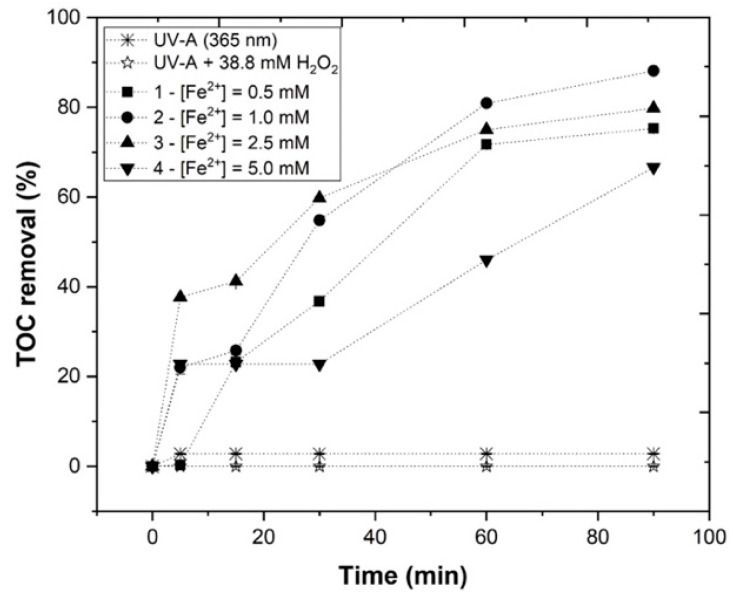
Progress of *TOC* percentage removal at different Fe^2+^ concentrations (0.5–5.0 mM) in the UV-A-Fenton process. Experimental conditions: [H_2_O_2_] = 38.8 mM, pH = 3.0, agitation = 350 rpm, *T* = 25 °C, radiation= UV-A, *I*_UV_ = 32.7 Wm^−2^, *t* = 90 min. Blank non-catalytic experiments (UV-A/H_2_O_2_ and UV-A without H_2_O_2_) are also represented as a reference.

**Figure 11 ijerph-19-04134-f011:**
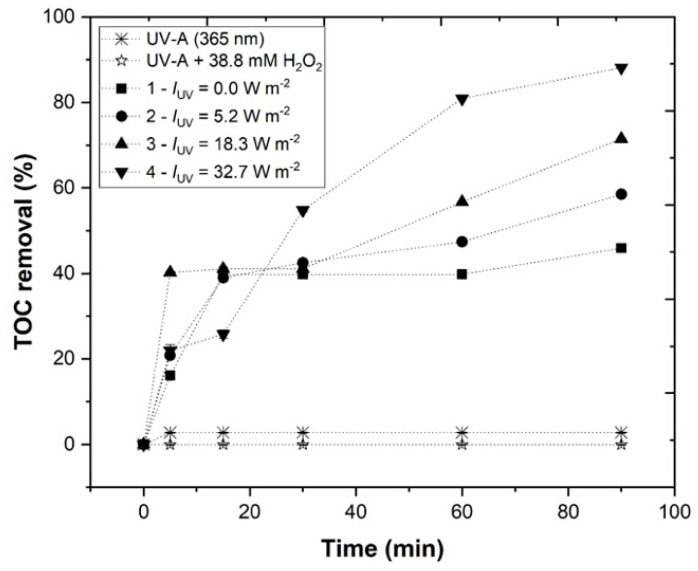
Progress of *TOC* percentage removal at different UV-A radiation intensities (0.0–32.7 W/m^2^) in the UV-A-Fenton process. Experimental conditions: [H_2_O_2_] = 38.8 mM, [Fe^2+^] = 1.0 mM, pH = 3.0, agitation = 350 rpm, *T* = 25 °C, *t* = 90 min. Blank non-catalytic experiments (UV-A/H_2_O_2_ and UV-A without H_2_O_2_) are also represented as a reference.

**Figure 12 ijerph-19-04134-f012:**
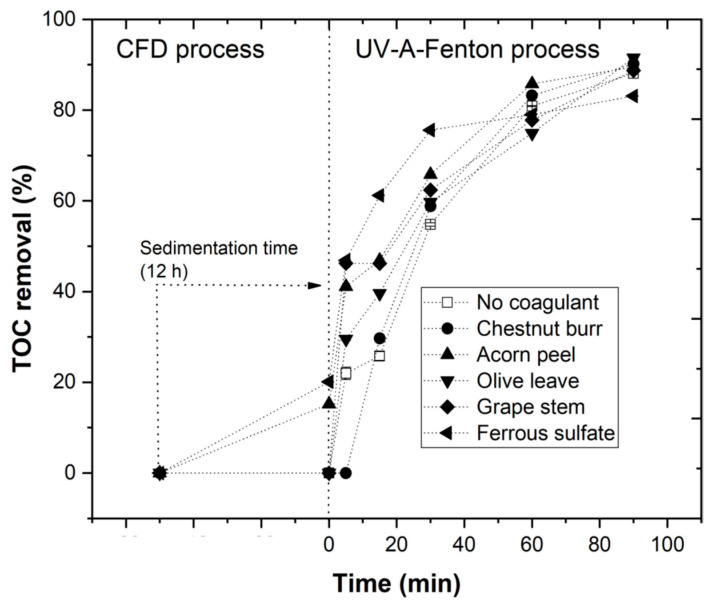
Progress of *TOC* percentage removal after CFD (with plant-based coagulants chestnut burr, acorn peel, olive leaf, and grape stem) and the UV-A-Fenton processes with the following experimental conditions: [H_2_O_2_] = 38.8 mM, [Fe^2+^] = 1.0 mM, pH = 3.0, agitation = 350 rpm, *T* = 25 °C, *t* = 90 min. Blank without coagulant and blank non-catalytic experiment (UV-A/H_2_O_2_) are also represented as a reference.

**Figure 13 ijerph-19-04134-f013:**
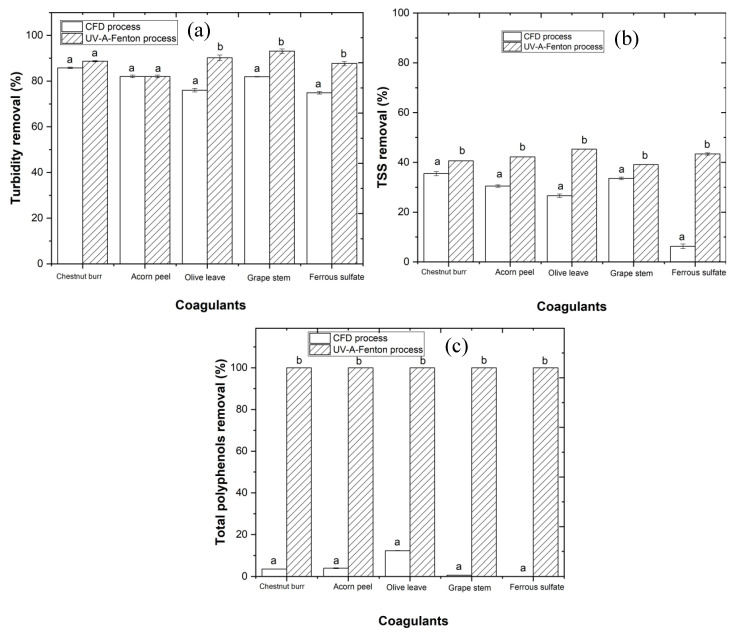
Progress of (**a**) turbidity, (**b**) TSS, and (**c**) total polyphenols percentage removal as a result of CFD process (with plant-based coagulants) in comparison with UV-A-Fenton under the following experimental conditions: [H_2_O_2_] = 38.8 mM, [Fe^2+^] = 1.0 mM, pH = 3.0, agitation = 350 rpm, *T* = 25 °C, *t* = 90 min. Different letters between the graphic bars mean values are statistically different (*p* < 0.05).

**Table 1 ijerph-19-04134-t001:** Elderberry wastewater characterization and legal limits for treated water release.

Parameters	Elderberry Wastewater	Portuguese Law Decree n° 236/98
pH	4.39 ± 0.04	6.0–9.0
Electrical conductivity (μS cm^−1^)	54.4 ± 10.2	
Turbidity (NTU)	16.0 ± 2.6	
Total suspended solids—TSS (mg L^−1^)	64.0 ± 2.5	60
Chemical Oxygen Demand—COD (mg O_2_ L^−1^)	773 ± 7.0	150
Biochemical Oxygen Demand—BOD_5_ (mg O_2_ L^−1^)	175 ± 18.0	40
BOD_5_/COD	0.23 ± 0.02	
Total Organic Carbon—*TOC* (mg C L^−1^)	144 ± 0.1	
Total polyphenols (mg gallic acid L^−1^)	37.8 ± 0.2	0.5

**Table 2 ijerph-19-04134-t002:** Plant-based coagulants sources: plant species, fruit common name, and re-used portion.

Plant Species	Fruit Common Name	Re-Used Portion
*Castanea sativa*	Chestnut	Burr
*Quercus ilex* and *Quercus rotundifolia*	Acorn	Peel
*Olea europaea*	Olive	Leaf
*Vitis vinifera*	Grape	Stem

**Table 3 ijerph-19-04134-t003:** Synergistic effect between photo-Fenton components.

Processes	k (min^−1^)		S (%)
H_2_O_2_ + UV-A	1.3 × 10^−3^ ± 0.4 × 10^−3^	a	
Fe^2+^ + UV-A	1.6 × 10^−3^ ± 0.3 × 10^−3^	a	
H_2_O_2_ + Fe^2+^ (Fenton)	6.4 × 10^−3^ ± 1.1 × 10^−3^	b	
H_2_O_2_ + Fe^2+^ + UV-A (5.2 W m^−2^)	8.0 × 10^−3^ ± 1.9 × 10^−3^	c	−11.3 ± 1.163 a
H_2_O_2_ + Fe^2+^ + UV-A (18.3 W m^−2^)	13.6 × 10^−3^ ± 0.9 × 10^−3^	d	36.4 ± 0.624 b
H_2_O_2_ + Fe^2+^ + UV-A (32.7 W m^−2^)	25.0 × 10^−3^ ± 1.5 × 10^−3^	e	64.4 ± 0.687 c

Different letters in the different rows mean that values are statistically different (*p* < 0.05).

## Data Availability

Not applicable.

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
