# Peer review of "Food By-Product Valorization by Using Plant-Based Coagulants Combined with AOPs for Agro-Industrial Wastewater Treatment"

_ijerph, 2022, doi:10.3390/ijerph19074134_

Round 1

Reviewer 1 Report

The authors studied 4 plant-derived coagulants obtained from chestnut burr, acorn peel, olive leaves, and grape stem to treat elderberry wastewater, varying the pH and coagulant dosage for use in combination with photo-Fenton treatment.  First of all, congratulations to the authors for their excellent work. However, they may consider to incorporate in the manuscript, the following suggestions:

  • In the introduction, it should be indicated in the main text the disadvantages of the use of aluminium and iron salts, compared to the proposed natural coagulants.
  • It should be indicated in the objectives the destination of the treated water.
  • Complete Table 1 with the last European directive approved on water quality.
  • Indicate the units of each of parameters or constants used in the equations. For this purpose, include a list with the nomenclature.
  • In terms of typographical issues, please revise the superscripts.
  • As a comparative table of final results, it can include with each of the coagulants (including aluminium salts) and the most favourable photo-Fenton process, the final results achieved in each of the parameters listed in the Portuguese Law Decree No 236/98, in order to clarify the contribution of the combination of processes.

Reviewer 3 Report

Article entitled Food by-products valorization as plant-based coagulants combined with AOPs for agro-industrial wastewater treatment written by Rita Beltrão Martins; Nuno Jorge; Marco S. Lucas; Anabela Raymundo; Ana I.R.N.A. Barros and José A. Peres and submitted to International Journal of Environmental Research and Public Health deals with an important issue of new materials usage for wastewater treatment.

The article is interesting and could be considered for publication in International Journal of Environmental Research and Public Health.

As English is not my native language, I am not able to assess language correctness.

However, while reading, I found some statements missing, confusing or unclear. Below I enclose the list of my comments.

All abbreviations, even as obvious as eg. FAO (line 39), should be explained at first appearance.

There is a lot of research available on the use of coagulants derived from natural organic materials. Although the Authors mention this in the Introduction, I think that a slightly larger review of the literature should be shown on the current state of knowledge on this subject. It is barely mentioned in the manuscript.

Lines 156 – 161: Sedimentation time was set as 12h. This is a lot compared to the sedimentation times for classic coagulants. Why was it decided for such a long time? What was the sedimentation rate?

Have any tests been carried out on the stability of the created organic coagulant? Various compounds can be leached out of it, leading to secondary pollution of wastewater. The treatment effect may therefore be related, on the one hand, to the removal of pollutants from the wastewater, and on the other hand, to their introduction from the coagulant.

When using solid coagulants, especially in the form of fine powders, sometimes there is a problem with dosing. The powders clump without dispersing in the volume of the solution. Due to the surface tension, they only float on the surface. Did the Authors encounter similar problems during the experiment, and if so, how were they solved?

Lines 256 -258 - repetition. Information is provided in detail in materials and methods.

What was TOC/COD dissolved? By comparing with the normally determined TOC and COD, the nature of the suspensions can be determined.

The Authors write that coagulation does not remove COD / TOC (line 264-265, Fig 5). It causes a slight removal of TSS and turbidity. However, the turbidity and TSS content is low, even very much compared to other effluents. So, what is the point of this process, especially since it is very time-consuming - 12 hours of sedimentation plus addition of reagents, mixing, etc.? The results, e.g., Figs 11, 12 indicate that there is no significant difference whether coagulants are used or not. Sometimes CFD even worsen the treatment effect. Starting with an initial TOC of 144mg / L of 5%, the difference in removal is very little difference.

How photo-Fenton process was terminated?

Hydrogen peroxide is well known COD determination disruptor. How the Authors solve this problem?

Was alkalization applied to obtain secondary coagulation in photo Fenton process?

The Authors write that they used a combined purification process, first CFD coagulation, then the photo-Fenton process. 4 coagulants are tested. On the other hand, I did not find the exact octane, after which the coagulant was applied, the wastewater was subjected to the Photo-Fenton process.

Fig 7: Usually, pH 3 is considered to be optimal for the photo-Fenton process. The higher the pH (closer to neutral), the worse the effectiveness. From the presented results it can be read that after 90min the effectiveness of the process at pH 7 is almost the same and higher than at 4 and 6. How do the Authors explain it, since it is rather inconsistent with the literature reports?

The content of tables 3 and 4 is incomprehensible to me. How is it possible that, for example, for the longest time, the observed is less peroxide consumption compared to the shorter times? At a pH of about 6, the lowest solubility of iron compounds is observed, and coagulation should take place. The results in the table absolutely do not indicate this. How do the Authors explain it?

Many parameters were determined in the raw wastewater. There is no information on how they changed after the applied treatment processes.

Raw coagulants were characterized in terms of material before the process. However, their characteristics after the process are missing.

Based on my comments and general impression I suggest major revision.

Round 2

Reviewer 2 Report

The revised manuscript submitted by Martins et al. has been improved compared to the former version. However, there are some still remarks and several changes to be done before accepting the manuscript for publication. Please see my comments below:

Line 26: Please replace [H2O2] with H2O2 and [Fe2+] with Fe2+. Please also replace it in the main text.

Line 18-34: The abstract was partly revised. However, there is still no justification why Photo-Fenton was chosen as APO. A short justification is enough. The statement that Photo-Fenton was applied to improve the global performance might be insufficient. Other APO could also improve the global performance. Maybe you had already experience with Photo-Fenton, or you knew that other APOs are less efficient?

Line 49/52: Please use a full stop between and write two separated sentences. Still, this read to complicated.

Line 96/101: Please add “organic” between “oxidizing” and “pollutant”

Line 112: Delete “near-“. (250 nm range is UV irradiation and not near-UV)

Line 128: I do not understand your argumentation stating that the TOC accuracy is higher than of COD and this is the reason why you have used TOC as major parameter for your investigation. Still, this needs to clarified properly.

Using the TOC as a sum parameter instead of sophisticated analyses such as LC/MS, GC/MS or others to determine the overall concentration of the sum of organic compounds of a sample is a rapid and reliable methods as long as the result is representative for the sample composition. Therefore, I agree using TOC as main parameter is a good choice instead of measuring different polyphenol fractions in each sample. And at the end, the conclusion is similar to the results of TOC measurements. Concerning the fact, that you may have surplus H2O2 disturbing your COD measurements you can stop the reaction either by using catalase addition or applying strong sonication. We found that the latter is more efficient and does not increase or disturb the COD or TOC measurement. Interfering Fe2+ can indeed cause mistakes in measuring COD. However, for Photo-Fenton little concentrations are practically used which do not strongly interfere with the COD reactant. Regarding your COD accuracy, you have about 99 % which is quite high and the measure range is also far from the limit of determination and detection. Thus, justifying based on accuracy is here misleading. But from a practical point of view, I understand that applying COD measurements with commercial test kits is costly compared with TOC measurements if the equipment is just available in your lab. And also, I guess that the TOC is more reliable to you because you want to investigate the decrease and removal of polyphenols. Therefore, using TOC instead of COD is justified.

Did you measure the COD and BOD5 of your final chosen treatment configuration? This could help to support data. If the TOC removal is 100% also the COD removal should be quite high, isn’t it?

And maybe you can give only a short statement at the beginning of section 3.3.1 indicating that using TOC measurement were appropriate for monitoring the removal of organic compounds representing polyphenols.

Line 196/203: Your concentration of H2O2 and Fe2+ is here give in Mol. However, later in the manuscript you report in the units mg/L. Please be consistent. I recommend reporting all in terms of “mg/L”. Please revise in the whole manuscript.

Line 355 and 332 in the old version of your manuscript:

Line 332: You mentioned that UV-A treatment was insufficient, but there is no further discussion about why. I guess that the reason it the monochromatic light source used. What was the reason to use this light source?

The reason why we have used UV-A in the present work was to test UV-LEDs with wavelength near visible light and because they are cheaper than UV-C LEDs and more environmentally friendly than UV-C mercury vapor lamps.

Your statement is very good and should be included in the main text at line 351 or similar.

Line 353: Why did you investigate pH 6 and 7? It must be expected that ferric iron precipitation must be expected lowering the reaction efficiency. Overall, this hole paragraph is too complicated written, and it is very tiring to read. Please revise and shorten.

Regarding Table S1, the H2O2 consumption was 49.0% at pH 3.0, decreasing to 41.3, 44.6 and 43.4%, respectively for pH 4.0, 6.0 and 7.0, meaning that a higher amount of  radicals were generated at pH 3.0. This higher H2O2 consumption can be explained by the iron speciation diagram as pH function. As observed in Table S2, the Fe2+ concentration after 90 min was 7.66, 5.40, 4.84 and 1.45 mg Fe L-1, respectively, for pH 3.0, 4.0, 6.0 and 7.0. Thus, increasing pH resulted in a shortage of which decreased the radicals availability in the medium, leading to iron Fe3+ precipitation as iron hydroxide Fe[OH]3, and consequently not reacting with H2O2, decreasing TOC removal [34,38,39].

I have seen little modification in line 377 to 384. However, the reason why you have performed different pH values with an expectable outcome is still not clarified. Did you want to determine the Fe specification? And if yes, why? Please explain this fact more detailed in section 3.2.2.

Line 442: Please justify why you assume a pseudo first-order reaction.

It was assumed a pseudo-first order reaction due to experimental data obtained. The hydroxyl radicals generated reacted almost instantly with organic compounds and we can considerer its concentration virtually constant. So, we have pseudo-first order reaction instead of second order.

Please add this information in 462 or similar.

Line 475: Your table must be revised. Please report k in scientific numbers including the standard deviation. Please use an extract column for your different letters indicating statistic differences. In addition, you can report the half-lives determined from your data fitting and either the sum of squared residuals or R2. The former should be more appropriate. Further, you should add your data fitting as graphs including original degradation curve and theoretical fitting curves with relevant data fitting in the supporting materials (Supplements).

Reviewer 3 Report

This is my second review of this article. The Authors answered all my questions and comments. All suggested corrections have been applied. I suggest to accept this article in its present form.

Author Response

Reviewer 3

"This is my second review of this article. The Authors answered all my questions and comments. All suggested corrections have been applied. I suggest to accept this article in its present form."

Authors would like to thank reviewer 3 for their useful comments that allowed to improve the final quality of the manuscript.  

Round 3

Reviewer 2 Report

The manuscript has been well improved and is now acceptable for publication. Congratulation